# Phosphorylation of Thymidylate Synthase and Dihydrofolate Reductase in Cancer Cells and the Effect of CK2α Silencing

**DOI:** 10.3390/ijms24033023

**Published:** 2023-02-03

**Authors:** Patrycja Wińska, Anna Sobiepanek, Katarzyna Pawlak, Monika Staniszewska, Joanna Cieśla

**Affiliations:** 1Chair of Drug and Cosmetics Biotechnology, Faculty of Chemistry, Warsaw University of Technology, 00-664 Warsaw, Poland; 2Chair of Analytical Chemistry, Faculty of Chemistry, Warsaw University of Technology, 00-664 Warsaw, Poland; 3Centre for Advanced Materials and Technologies, Warsaw University of Technology, Poleczki 19, 02-822 Warsaw, Poland

**Keywords:** thymidylate synthase TS, dihydrofolate reductase DHFR, serine hydroxymethyltransferase SHMT, acute lymphoblastic leukaemia, lung adenocarcinoma, protein phosphorylation, protein kinase CK2, in-gel kinase assay

## Abstract

Our previous research suggests an important regulatory role of CK2-mediated phosphorylation of enzymes involved in the thymidylate biosynthesis cycle, i.e., thymidylate synthase (TS), dihydrofolate reductase (DHFR), and serine hydroxymethyltransferase (SHMT). The aim of this study was to show whether silencing of the CK2α gene affects TS and DHFR expression in A-549 cells. Additionally, we attempted to identify the endogenous kinases that phosphorylate TS and DHFR in CCRF-CEM and A-549 cells. We used immunodetection, immunofluorescence/confocal analyses, reverse transcription–quantitative polymerase chain reaction (RT-qPCR), in-gel kinase assay, and mass spectrometry analysis. Our results demonstrate that silencing of the CK2α gene in lung adenocarcinoma cells significantly increases both TS and DHFR expression and affects their cellular distribution. Additionally, we show for the first time that both TS and DHFR are very likely phosphorylated by endogenous CK2 in two types of cancer cells, i.e., acute lymphoblastic leukaemia and lung adenocarcinoma. Moreover, our studies indicate that DHFR is phosphorylated intracellularly by CK2 to a greater extent in leukaemia cells than in lung adenocarcinoma cells. Interestingly, in-gel kinase assay results indicate that the CK2α’ isoform was more active than the CK2α subunit. Our results confirm the previous studies concerning the physiological relevance of CK2-mediated phosphorylation of TS and DHFR.

## 1. Introduction

One of the best-known post-translation modifications is protein phosphorylation, which mainly occurs at the serine, threonine, and tyrosine residues of protein sequences. These amino acids’ phosphorylation status may directly or indirectly regulate protein functions. The phosphorylation process can be described as binding a protein kinase to a distinct protein associated with the transfer of phosphate group from ATP and releasing of ADP and phosphorylated protein. This phosphorylation reaction is usually transient, and protein phosphatases carry out dephosphorylation. Protein–protein interactions are mediated by phosphor-binding domains created by specific phosphorylation sites [1]. The fundamental mechanism of cellular signal transduction involves protein phosphorylation that allosterically changes its activity and affects the formation of protein complexes [2,3]. Over the past decades, many kinases have been associated with cancer biology; most of them are tyrosine-specific kinases. Therefore, protein kinase inhibitors emerged as a new drug class for cancer treatment [4].

Among 518 protein kinases identified in the human genome, protein kinase CK2 (CK2; formerly known as casein kinase II, EC 2.7.11.1) has gained considerable interest. CK2 is a constitutively active serine/threonine kinase. Its activity is regulated by protein–protein interactions, changes in its intracellular concentration, level of the phosphorylation state, oligomerisation, and cellular localisation [5]. An increased level/activity of CK2 kinase is observed in several tumour types [6,7]. CK2 kinase plays an important role in cancer transformation, and it was demonstrated that cancer cells are more sensitive than normal cells to the inhibition of CK2 activity. High levels of CK2 are associated with poor prognosis in multiple cancer types, and therefore, the enzyme is intensively studied as a target for cancer therapy [8].

Approximately one-third of CK2 substrates, including transcription factors, are involved in gene expression and protein synthesis. Several CK2 substrates are classical metabolic enzymes [9]. These include thymidylate biosynthesis enzymes. The de novo biosynthesis of 2’-deoxythymidine-5’-monophosphate (thymidylate, dTMP) (Figure 1) is catalysed by the enzyme thymidylate synthase (TS, EC 2.1.1.45). In this reaction, the methylene residue is transferred from N5,10-methylenetetrahydrofolate (meTHF) to dUMP, resulting in 7,8-dihydrofolate (DHF) and dTMP. In a subsequent reaction catalysed by dihydrofolate reductase (DHFR, EC 1.5.1.3), DHF is reduced to tetrahydrofolate (THF), a substrate for serine hydroxymethyltransferase (SHMT, EC 2.1.2.1). In SHMT-catalysed reactions, meTHF is regenerated with the concomitant conversion of serine into glycine. Research by Prof. Rode’s team has shown that recombinant thymidylate synthase (TS) is phosphorylated by CK2 [10]. The studies conducted with human TS revealed that decreased Vmax app values characterised its CK2α-phosphorylated form without affecting Km values. The results of molecular dynamics simulations have shown that CK2-mediated phosphorylation of serine 124 residue of the human TS leads to a protein conformational change, resulting in an unfavourable position of the substrate (dUMP) and cofactor (meTHF) in the active centre [11]. In addition, a stiffening of specific protein fragments, especially the loop closing the active centre pocket, has been shown [10]. High-resolution structures have provided evidence that the native human TS can have two major conformations: active and inactive, which depend on the loop’s location 182–297 [12]. Mutant TS, M190K, with loop 182–297 stabilised in an inactive conformation, was highly phosphorylated by CK2 in contrast to active-conformation-stabilised mutant R163K [13]. These data may indicate the physiological relevance of conformational switching of TS activity with possible stabilisation of the inactive form by phosphorylation. In vitro phosphorylation by CK2 of the second enzyme of the thymidylate synthesis cycle, DHFR, was recently demonstrated by our team [14].

We have shown for the first time that DHFR is in vitro phosphorylated on Ser168 and, to a lesser extent, Ser145, by the α subunit of CK2. Our other most recent research [15,16] indicated that CK2 inhibition does not change the levels of the transcripts; however, it affects the protein levels of DHFR and TS in both tested A549 and CCRF-CEM cell lines, and the cytosolic SHMT1 level in CCRF-CEM cells. Moreover, we have shown that CK2-mediated phosphorylation enables the interaction of phosphorylated TS (pTS) with SHMT1 and enhances the stability of the tri-complex containing SHMT1, DHFR, and pTS [16]. Our previous results suggest an important regulatory role of CK2-mediated phosphorylation for inter- and intracellular protein levels of enzymes involved in the thymidylate biosynthesis cycle. Taking into account that TS, DHFR, and CK2 are molecular targets in cancer chemotherapies, we have recently undertaken studies on the simultaneous treatment of cancer cells with inhibitors of TS and CK2α [17,18] or DHFR and CK2α [15] searching for a synergistic effect. The obtained results demonstrate the ability of CK2 inhibitors to enhance the efficacy of 5-FU or MTX in cancer cells.

This work aimed to elucidate whether endogenous protein kinase CK2 phosphorylates TS and DHFR in leukemic (CCRF-CEM) and lung adenocarcinoma (A-549) cells. We also investigated the cellular expression and localisation of TS and DHFR proteins in CK2α-deficient A-549 cells obtained by the silencing of the *CSNK2A1* gene with siRNA.

## 2. Results

### 2.1. Protein Levels of TS and DHFR in A-549 with the Silenced CSNK2A1 Gene

To confirm our previous results indicating the physiological role of CK2-mediated phosphorylation of the enzymes involved in thymidylate synthesis cycle [16], we performed silencing of the CK2-encoding gene in the A-549 cell line (lung adenocarcinoma) with small interfering RNA. We treated A-549 cells with 25 nM, 50 nM, and 100 nM *CSNK2A1* siRNA for 48 h and 72 h. Western blot analysis revealed significant silencing of the gene expression (Figure 2a,b,e). We observed that the level of TS and DHFR in control cells, which were not treated with siRNA, dramatically decreased after 72 h in comparison to 48 h of culture (Figure 2a), which was expected [16]. After 48 h of treatment with *CSNK2A1* siRNA, both TS and DHFR levels did not change significantly with the increasing *CSNK2A1* siRNA concentration (Figure 2a,b). However, after 72 h of treatment, we observed that the levels of both enzymes increased with increasing siRNA concentration up to 270-fold and 21-fold higher than in control cells, respectively (Figure 2a,c,d).

### 2.2. TS and DHFR mRNA Levels in A-549 with the Silenced CSNK2A1

To explain the changes in the DHFR and TS protein levels in cells treated with *CSNK2A1* siRNA, the mRNA levels of TS, DHFR, and CK2α were tested using the Q-PCR method, and the relative gene expression is shown in Figure 3. A significant decrease in CK2α transcript was detected in A-549 treated with *CSNK2A1* siRNA with the relative gene expression corresponding to 25% (100 nM siRNA) and 29% (50 nM siRNA) of control after 48 h incubation (Figure 3a) and 19% (100 nM siRNA), 24% (50 nM siRNA), and 31% (25 nM siRNA) of control after 72 h of incubation (Figure 3b). The relative expression of TS gene (*TYMS*) significantly increased in the cells treated with 50 nM and 100 nM *CSNK2A1* siRNA, with the highest value of 419% of control signal detected after 72 h of incubation with 50 nM *CSNK2A1* siRNA (Figure 3b). The relative expression of *DHFR* was also increased in cells treated with *CSNK2A1* siRNA, with the highest value of 851% of control detected in cells treated with 50 nM *CSNK2A1* siRNA for 72 h (Figure 3b). The obtained results correspond well with the changes in the TS and DHFR protein levels, suggesting that the increase in protein levels of TS and DHFR may be caused by an increase in the amount of their transcripts.

### 2.3. The Effect of Silencing of CSNK2A1 on Localisation of DHFR and TS in A-549 Cells

To explain the regulatory role of presumed CK2-mediated phosphorylation of DHFR and TS, we investigated the effect of *CSNK2A1* silencing on the distribution and localisation of DHFR and TS in A-549 cells. Immunofluorescence was used to detect TS, DHFR, and additionally CK2α in A-549 cells after 72 h treatment with 100 nM *CSNK2A1* siRNA, and the results are shown in Figure 4. The comparison of the distribution of investigated thymidylate synthesis cycle enzymes in the untreated lung cancer cells shows that both TS and DHFR are almost uniformly distributed in the cells (in the cytoplasm and nuclei, Figure 4a,b). At the same time, DHFR in the cytosol locates closer to the nuclei than TS (Figure 4b).

Upon treating cells with *CSNK2A1* siRNA, both the localisation and the levels of TS and DHFR changed, with a higher signal emitted by the enzymes in the cytosol than in the nuclei. This manifests itself by a stronger fluorescence of nuclei as a consequence of a reduced green fluorescence signal corresponding to TS and DHFR levels (Figure 4d,e). Additionally, the obtained results for CK2α confirm its significant decrease in cells treated with *CSNK2A1* siRNA (Figure 4f). The obtained results are consistent with the Western blot results, indicating a significant increase in TS and DHFR levels in A-549 treated with 100 nM CSNK2A1 siRNA. However, it should be taken into account that immunofluorescence, in contrast to Western blot analysis, allows the detection of the attached cells only.

### 2.4. Phosphorylation of TS and DHFR by Endogenous Protein Kinases from CCRF-CEM and A-549 Cells

We used in-gel kinase assay and mass spectrometry analysis to detect cellular protein kinases phosphorylating TS and DHFR in CCRF-CEM (acute lymphoblastic leukaemia) and A-549 cells (lung adenocarcinoma). Each in-gel kinase assay was performed in parallel for the gels without substrate (control gels) and with the co-polymerised substrate, i.e., recombinant TS or recombinant DHFR (Appendix A), to exclude that the observed results were due to autophosphorylation of protein kinases. The results are shown in Figure 5. We have observed that some parts of co-polymerised enzymes migrate during SDS-PAGE according to their molecular weight, i.e., 36 kDa and 21 kDa for TS and DHFR, respectively (Appendix A). We have also noticed that co-polymerisation of DHFR was better at an acrylamide:bis-acrylamide ratio of 19:1 than 37.5:1 (not shown). As a positive control of CK2-mediated phosphorylation of TS and DHFR, we used two recombinant catalytic subunits of CK2, i.e., α and α’. We detected phosphorylation of both TS and DHFR by recombinant CK2α’ and by endogenous CK2 (Figure 5a,c). We also noted that DHFR is phosphorylated by endogenous CK2 to a greater extent than TS (Figure 5c), so we had to shorten the film exposure time. Moreover, the level of DHFR phosphorylation was significantly higher in leukemic cells (CCRF-CEM) than in the lung adenocarcinoma cell line (A-549) and corresponded to increased autophosphorylation of kinases in leukemic cells (Figure 5d).

In contrast to DHFR, we did not observe significant differences between CK2-mediated phosphorylation of TS in leukemic and lung adenocarcinoma cells. However, the extent of this phosphorylation was lower than for DHFR. We have also concluded that CKα’ undergoes much stronger autophosphorylation than the CK2α subunit. In addition to CK2-mediated phosphorylation of TS and DHFR, strong kinase activity corresponding to the molecular mass of 55 kDa was detected using an in-gel kinase assay (Figure 5a,c). Moreover, this kinase was highly autophosphorylated in both cell lines (Figure 5b,d).

### 2.5. Confirmation of the Identity of Casein Kinases 2 Subunit α and α

Mass spectrometry analysis was used to confirm whether endogenous CK2α and its isoform CK2α’ are responsible for the phosphorylation of TS/DHFR. Molecular weight, Score, and emPAI parameters for identified proteins were used to describe the size distribution of proteins in the excised gel band. The relative spectral amount of protein was established as the ratio of emPAI parameter obtained for protein against the sum of the emPAI values (total protein content, TPC) for all 185/180 identified proteins. Based on the results, it can be concluded that the proteins in terms of size in the excised gel band are homogeneous (43.4 ± 1.8 kDa), representing 60% of TPC (Figure 6). Actins (four of the most abundant proteins) and the twelve types of kinases represent 15% and 21% of TPC in the gel band. This result reflects the disproportion of protein content in biological samples. It also shows that the proteins being searched for should, with high probability, be present in the gel band being tested. The expected casein kinase (CK2α) and its isoform (CKα’) in the analysed gel band represent 0.8% and 0.3% of TPC, respectively. Their identity was established with considerable confidence—significant Scores of 559 and 218 (Appendix A), indicating good agreement for monoisotopic masses of detected peptides and their fragments (Appendix A). It should be noted that all detected peptides based on which proteins were indicated are unique to this protein family, which was verified using the UniProtKB database. This significantly increases the veracity of the results obtained.

### 2.6. Identification of Other Protein Kinases

Twelve kinases were detected in a gel band corresponding to the molecular mass of 55 kDa, seven of which were detected as phosphorylated. At the same time, only two proteins (serine/threonine protein kinase receptor, STKR, and activin receptor type-1C, AR1C) were spectrally defined with sufficient accuracy, and their monoisotopic mass was within the range of the molecular mass defined for the gel band (Appendix A). Interestingly, activin receptor type-1C is a serine/threonine protein kinase, forming a receptor complex on ligand binding [19].

## 3. Discussion

To investigate the physiological role of CK2-mediated phosphorylation of TS and DHFR, we tested the effect of silencing of the *CSNK2A1* gene, encoding the main catalytic subunit of CK2, i.e., CK2α, on both transcripts and protein levels of TS and DHFR in A-549 cells. Various reports indicate the occurrence of CK2-mediated phosphorylation of TS and DHFR. However, to date, no studies have shown the effect of *CSNK2A1* silencing on these enzymes. The significant decrease in TS and DHFR protein levels in untreated control cells after 72 h compared to 48 h culture agrees well with our previous results. It corresponds to the exit of the cells from the cell cycle S-phase [16]. Our results also agree with the literature data showing that the TS mRNA level increases up to 20-fold in the S-phase [20], which is driven by the E2F transcription factor [21]. In contrast, the transcription of DHFR in the S-phase is driven by the E2F transcription factor working in concert with the Sp1 transcription factor [22].

The data indicating a significant increase in DHFR and TS proteins in A-549 cells with CK2 deficiency corresponds with our previous studies on both A-549 and CCRF-CEM in which we used CX-4945 to selectively inhibit CK2 [16]. However, contrary to our previous results showing no significant changes in DHFR and TS transcripts after CK2 activity inhibition [15], a substantial increase in both TS and DHFR transcripts was detected in A-549 cells treated with *CSNK2A1* siRNA. Thus, the presented RT-QPCR results correspond better to the changes in the protein level of both TS and DHFR, and consequently suggest an increase in their gene expression in cells with a deficiency of CK2α activity. There is much evidence that CK2 regulates gene expression directly or indirectly [23]. The studies on leukaemia cells have indicated that phosphorylation of Ikaros impaired the protein’s ability to regulate both the transcription of its target genes and the global epigenetic landscape in leukaemia. Moreover, our results obtained for TS are in agreement with the previous studies carried out on recombinant forms of TS [24], showing that phosphorylated, compared to non-phosphorylated, recombinant enzyme forms bind their cognate mRNA (tested only with the rat enzyme) and repress their mRNA translation (tested with human, rat, and mouse enzyme). Thus, the lack of phosphorylation of the TS protein consequently leads to increased transcript and protein levels in the cell.

In addition, our present results indicating the effect of CK2α silencing on the distribution of both TS and DHFR in A-549 cells are in agreement with our previous studies obtained for A-549 cells after treatment with CX-4945, a selective inhibitor of CK2 [16]. The results suggest that CK2-mediated phosphorylation can affect their cellular localisation and could be necessary for nuclear localisation. It is well established that post-translational protein modifications are crucial in many processes. Stover’s group provided evidence that all three enzymes of the thymidylate synthesis cycle undergo SUMOylation, which provides a mechanism by which they are directed and compartmentalised in the nucleus [25,26,27]. The relationship between SUMOylation/ubiquitination and phosphorylation has not been investigated in the case of thymidylate synthesis enzymes. However, the “cross-talk” between phosphorylation and SUMOylation/ubiquitination is widely appreciated [28,29,30], and there are numerous examples of the connection between SUMO system and CK2 signalling [31]. Such interdependence has been recently shown; for example, (i) cytomegalovirus transactivator IE2 CK2-mediated phosphorylation increased its SUMO-dependent activity [32]; (ii) the stability of the Sensitive to Apoptosis Gene (SAG), a RING component of SCF E3 ubiquitin ligase, was regulated by CK2-mediated phosphorylation on Thr10 [33]; (iii) CK2 has been identified as a new important modulator of the stability of homeodomain protein PDX-1 [32]; (iv) CK2α phosphorylated Ser154 in Nur77 and thereby regulated Nur77 protein levels by promoting its ubiquitin-mediated degradation [34]. Interestingly, phosphorylation was found to also mark TS for proteolytic degradation by the ubiquitin–proteasome system [35], although the involved protein kinase has not yet been identified.

In this paper, we show for the first time that both TS and DHFR are likely phosphorylated by endogenous CK2 in two types of cancer cells, i.e., acute lymphoblastic leukaemia (CCRF-CEM) and lung adenocarcinoma (A-549). We have also demonstrated that the level of DHFR phosphorylation is significantly higher in CCRF-CEM than in A-549. These results correlate well with the literature data demonstrating that CK2 activity is upregulated in various types of cancer, including acute lymphoblastic leukaemia [7,23].

Although previous results with recombinant proteins have shown that both recombinant TS [10] and recombinant DHFR [14] are phosphorylated by CK2α, our in-gel kinase assay results indicate that both TS and DHFR are better phosphorylated by the CK2α’ isoform than by CK2α. Since during in-gel kinase assay, proteins are renatured after SDS-mediated denaturation, the different capacities of individual kinases for renaturation and recovery of their enzymatic activity must be considered [36]. The in-gel kinase assay results also indicate strong autophosphorylation of both recombinant isoforms of CK2, which agrees with the literature data [37]. It was shown that CK2 can self-regulate its activity through autophosphorylation, which can occur either in free catalytic subunits or within the holoenzyme. Within the free catalytic subunits, this occurs intermolecularly at Y182 on CK2α and Y183 on CK2α’, resulting in increased catalytic activity through modulating interactions between the residue and the N-terminal of the kinase. Interestingly, the studies indicated that holoenzyme formation inhibits this autophosphorylation, so it can only occur in catalytic subunits not complexed with CK2β [38].

Moreover, in addition to CK2-mediated phosphorylation of TS and DHFR, significant kinase activity, detected using in-gel assay, can be attributed to a protein with a molecular weight of 55 kDa. Interestingly, the studies indicate that this kinase is highly autophosphorylated. Among seven kinases indicated by LC-MS as phosphorylated, only two protein kinases, i.e., transmembrane receptor protein serine/threonine kinase (ID: A0A0B5HR54) and activin receptor type-IC, serine/threonine protein kinase (ID: Q8NER5), are spectrally defined with sufficient accuracy. Such results do not preclude phosphorylation on other kinases but indicate that these two proteins were the most abundant. Interestingly, activin receptor type-IC is serine/threonine protein kinase, forming a receptor complex on ligand binding. The receptor complex consists of transmembrane serine/threonine kinases type II and type I. Type II receptors phosphorylate and activate type I receptors, which undergo autophosphorylation, then bind and activate SMAD transcriptional regulators, which subsequently translocate to the nucleus and interact directly with DNA alone or as its complex with other transcription factors [19]. The presented results constitute the first preliminary evidence that TS and DHFR may be phosphorylated not only by CK2 but also by another serine/threonine kinase. However, these data require further in-depth research.

Considering all the available data, it seems likely that CK2-mediated phosphorylation of thymidylate synthesis enzymes regulates in vivo their levels, catalytic activities, intracellular protein complex formation, and cellular distribution. However, it cannot be excluded that other kinases may be involved in the phosphorylation of TS and DHFR, which requires further research. The phosphorylation of the third thymidylate synthesis cycle enzyme (SHMT) is probable, as quantitative proteomic studies have shown that its cytosolic (SHMT1) and mitochondrial (SHMT2) forms are substrates for CK2α [39]. Furthermore, our preliminary results obtained with in-gel kinase assay agree with these data (not shown). Moreover, the status of thymidylate synthesis cycle enzymes phosphorylation may affect the outcome of cancer treatment. Therefore, elucidating the complex physiological role of the effect of CK2-mediated phosphorylation on those enzymes that have not been investigated so far is of great importance.

## 4. Materials and Methods

### 4.1. Reagents and Antibodies

pET28a vector was purchased from Novagen (Madison, WI, USA). ON-TARGETplus Human *CSNK2A1* (1457) siRNA—SMARTpool, 5 nmol/L, DharmaFECT 1 siRNA Transfection Reagent, and other reagents used in silencing the CK2α gene were purchased from Horizon Discovery (Waterbeach, UK). [γ-32P] ATP was purchased from Hartmann Analytic (Braunschweig, Germany). NiNTA-Agarose was from Qiagen (Venlo, Netherlands). X-ray films and autoradiography reagents were from Primax (Poznań, Poland). RIPA buffer was from Cell Signaling Technology (Danvers, MA, USA). Protease inhibitor cocktail was from Thermo Scientific (Waltham, MA, USA). The following primary antibodies were used: anti-GAPDH #MAB374, 1:500 (Merck Millipore, Darmstadt, Germany), anti-DHFR (BD Biosciences, Franklin Lakes, NJ, USA) and anti-TS #MAB4130, 1:500 (Merck Millipore, Darmstadt, Germany), anti-DHFR #sc-377091, 1:100 (Santa Cruz Biotechnology, Dallas, TX, USA), anti-TS # sc-390945, 1:100 (Santa Cruz Biotechnology, Dallas, TX, USA). Secondary antibodies were goat anti-rabbit IgG-HRP #P0448, 1:2000, 1h, RT (Dako, Santa Clara, CA, USA) and anti-mouse IgG-HRP #P0447, 1:1000, 1h, RT (Dako, Santa Clara, CA, USA). Hoechst 33,342 #H3569 (Life Technologies, Carlsbad, CA, USA) and Goat anti-Mouse IgG (H + L) Cross-Adsorbed Secondary Antibody, Alexa Fluor™ 488 (green fluorescence) were used in the IF study. Protease inhibitors (#11 836 153 001) were from Roche Applied Science (Mannheim, Germany). Nitrocellulose membrane was from GE Healthcare Life Sciences (Freiburg, Germany) and ECL reagent was from Biorad (Hercules, CA, USA). Other solvents, reagents, and chemicals were purchased from POCH (Avantor Performance Materials, Gliwice, Poland), Merck, and Sigma-Aldrich Chemical Company (St. Louis, MO, USA).

### 4.2. Cell Culture

CCRF-CEM (ECACC 85112105) human Caucasian acute lymphoblastic leukaemia was purchased from ECACC, and A-549 (ATCC CCL 185) human lung carcinoma was purchased from ATCC. The A-549 cell line was cultured in F-12K Medium (Kaighn’s Modification of Ham’s F-12 Medium, Gibco, Waltham, MA, USA) supplemented with 10% foetal bovine serum (Gibco, Waltham, MA, USA), 2 mM L-glutamine, and antibiotics (100 U/mL penicillin, 100 µg/mL streptomycin). CCRF-CEM was cultured in RPMI 1640 supplemented with 10% foetal bovine serum (Sigma-Aldrich, St. Louis, MO, USA), 2 mM L-glutamine, and antibiotics 100 U/mL penicillin, 100 µg/mL streptomycin (Sigma-Aldrich, St. Louis, MO, USA). Cells were grown in 75 cm^2^ cell culture flasks (Sarstedt, Nümbrecht, Germany) in a humidified atmosphere of CO_2_/air (5/95%) at 37 °C. All the experiments were performed in exponentially growing cultures.

### 4.3. Protein Preparations

hTS-N-HisTag/pET28a/BL21(DE3) and hDHFR-N-HisTag/pET28a/BL21(DE3) were obtained as described previously [14,40]. Recombinant His-tagged DHFR and TS were overexpressed and purified as described previously [14,40]. CK2α and α’ were obtained by cloning using standard procedures described by us previously [41]. Protein concentration was assayed using the Bradford method [42].

### 4.4. In-Gel Kinase Assay

In-gel kinase assay was performed according to protocols described elsewhere [36,43]. Human recombinant TS or DHFR was co-polymerised in polyacrylamide gel. After SDS-PAGE of tested samples (human recombinant CK2α, CK2α’, and cellular lysates of CCRF-CEM and A-549), SDS was removed (25 mM Tris-HCl pH 7.5; 5 mM NaF; 0.1 mM Na3VO4; 0.05% BSA; 0.1% Triton X-100; 0.1 mM DTT), and following protein renaturation (25 mM Tris-HCl pH 7.5; 5 mM NaF; 0.1 mM Na3VO4; 0.1 mM DTT), the in-gel reaction for protein kinase was performed with radiolabelled gamma-P32-ATP in a final volume of 30 mL of a reaction buffer (150 µM ATP; 40 mM HEPES pH 7.4; 13,5 mM MgCl2; 2 mM EGTA pH 7; 0.1 mM Na_3_VO_4_; 0.1 mM DTT) with gentle stirring (50 rpm) at 30 °C for 90–105 min. The kinase activity was detected using the autoradiography method. To make sure that the observed results are not the consequence of the autophosphorylation, in-gel kinase assay was performed in parallel gels: with and without substrate, the latter being the control gel.

### 4.5. Mass Spectrometry Analysis

The separated gel band (corresponding to proteins with molecular weight ~40 kDa and ~55 kDa) after resolving proteins in a sample by SDS-PAGE and staining with Coomassie Brilliant Blue was subjected to LC-MS/MS analysis after digestion with trypsin.

#### 4.5.1. Large-Scale Mass Spectrometry Analysis

Amino acid sequences were established for 1423 (exp1) and 2207 (exp2) ions with a Score above 37 for individual ions and extensive homology (*p* < 0.00169). High Scores were achieved by following the appropriate criteria: (1) the allowable difference between the MS-determined monoisotopic mass and calculated one could not exceed ± 5 ppm, (2) the allowable monoisotopic mass difference for fragment ions could not exceed ± 0.01 Da. A minimum of 2 peptides were required to establish the identity of a protein unless its sequence was unique to a family of proteins with an established Homo Sapiens taxonomy; then, one peptide was considered sufficient [44].

#### 4.5.2. Evaluation of Mass Spectrometry Platform

Evaluation of the qualitative analysis using the MASCOT platform (date stamp 31 February 2022) was performed by determining the identity of the peptides against Decoy Database Searching containing reversed peptide sequences [45,46]. The false discovery rate (FDR) was below 1%. Sequences of 13/20 “decoy” peptides were not present in the identified proteins.

#### 4.5.3. Size Distribution of Proteins in Excised Gel Band and Their Relative Concentration

MASCOT, from the MS/MS data, assigns a specific protein with a known molecular weight and determines the probability of a random match (Score) by assessing the degree of match between the monoisotopic mass of the peptides and their fragments. The magnitude of this parameter will depend on the protein concentration, the accuracy of the MS/MS method, and the purity of the sample—assigned proteins are listed by Score (Appendix A). Another parameter has been introduced to conduct relative (label-free) quantification of the proteins in a mixture—The Exponentially Modified Protein Abundance Index (emPAI) [47]. It is based on protein coverage by the peptide matches in a database search result, which depends on the protein’s concentration.

### 4.6. Obtaining A-549 Cell Line with the Silenced CSNK2A1 Gene

CK2α subunit gene (*CSNK2A1*) was silenced with small interfering RNA directed against the appropriate mRNA sequence using ON-TARGETplus according to the procedure described [48] and the manufacturer’s recommendations (https://horizondiscovery.com/en/gene-modulation/knockdown/sirna/on-targetplus-sirna (accessed on 3 January 2023)). siRNA was resuspended in RNase-free 1x siRNA Buffer for the desired final concentration using volumes listed in the table by the manufacturer (siRNA resuspension protocol). The concentration of resuspended siRNA was quantified by measuring of absorbance at 260 nm (A260) with the use of a dual-beam spectrophotometer. RNA was used immediately or aliquoted into smaller volumes to limit the number of freeze–thaw cycles. For cell transfection, the general protocol for the use of DharmaFECT™ transfection reagents to deliver siRNA into cultured mammalian cells was used (DharmaFECT™ Transfection Reagents—siRNA transfection protocol).

### 4.7. Western Blotting

A-549 cells growing exponentially were seeded at 10^5^ cells/mL in 6 cm diameter plates in 5 mL, whereas CCRF-CEM cells were seeded in the concentration of 2 × 10^5^ cells/mL in 25 cm^2^ flasks in 10 mL (Sarstedt). After incubation, suspension cells were collected by centrifugation at 260 RCF, washed 3x with ice-cold PBS, supernatants were discarded, and pellets were stored at –20 °C for up to one month. Adherent monolayer cells were washed three times in ice-cold PBS. The cells were scraped and lysed in RIPA (50 mM Tris-HCl pH 7.4, 1% NP-40, 0.5% sodium deoxycholate, 0.1% SDS, 150 mM NaCl, 2 mM EDTA, 50 mM NaF, 0.2 mM sodium orthovanadate, and protease inhibitors cocktail) as was described previously [16].

### 4.8. Immunocytochemical Staining and Microscopy Analysis

A-549 cells were seeded in 4-well dishes (35/10 mm; Greiner Bio-One North America, Inc., Monroe, NC, USA). and cultured under appropriate conditions. After 18 h of culturing, cells were treated with 100 nM *CSNK2A1* siRNA. Subsequently, cells were washed twice with PBS, fixed with 3.7% paraformaldehyde solution (PFA) for 20 min, washed twice with PBS, and incubated for 30 min at room temperature with permeabilisation and blocking solution (5% BSA, 0.1% Triton X 100 and 0.5% Tween 20 in PBS). Subsequently, a solution was discarded, and cells were washed three times with PBST (0.1% Tween 20 in PBS) and incubated with the primary antibodies (anti-DHFR or anti-TS, both 1:100) in PBST for 24 h at 4 °C. After washing with PBST (0.5 mL/compartment), cells were protected from light and incubated with Alexa Fluor 488-conjugated anti-mouse diluted 1:500 (Thermo Fisher Scientific, USA) for 1 h at room temperature. After washing, cells were incubated in 1 µL/mL Hoechst 33,342 in PBST for 15 min. The dye solution was discarded, the cells were washed with PBST (0.5 mL/compartment), and a drop of microscope slide mounting medium (Roth) was added. Microscopic observations and images were performed using the Zeiss Axio Observer 7 LSM 900 confocal laser scanning microscope equipped with Airyscan 2 detector and ZEN software (Zeiss, Oberkochen, Germany).

### 4.9. RNA Isolation Followed by the Reverse Transcription–Quantitative Polymerase Chain Reaction (RT-qPCR)

The total RNA was extracted using RNA extracol (EurX, Gdansk, Poland) according to the manufacturer’s instructions. RNA quality was checked on the 1% agarose gel and the quantity of RNA on the NanoDrop One/One UV-Vis Spectrophotometer (Thermo Fisher Scientific, Waltham, MA, USA). Next, 1 µg of RNA was reverse transcribed into cDNA using the RevertAid First Strand cDNA Synthesis Kit (Thermo Fisher Scientific, Waltham, MA, USA) according to the producer’s instructions and qPCR was performed using the RT PCR Mix SYBR^®^ (A&A Biotechnology, Gdańsk, Poland) on the CFX Opus Real-Time PCR Systems (Bio-Rad, Hercules, California, USA). The following primers were used: target gene TYMS (Fw 5′-TGGAATCCAAGAGATCTTCCTC-3′; Rv 5′-TTCAGGCCCGTGATGTG-3′), target gene DHFR (Fw 5′-GCGTTCTGCTGTAACGAG-3′; Rv 5′-ACCAGATTCTGTTTACCTTCTAC-3′), target gene CSNK2A1 (Fw 5′-GGTGAGGATAGCCAAGGTTCTG-3′; Rv 5′-TCACTGTGGACAAAGCGTTCCC-3′) and reference gene GAPDH (Fw 5′-AGGGCTGCTTTTAACTCTGGT-3′; Rv 5′-CCCCACTTGATTTTGGAGGGA-3′). The PCR cycling conditions were set as follows: initial denaturation for 3 min at 95 °C, and then 40 cycles of 15 s at 95 °C, 60 s at 55 °C and 45 s at 72 °C. The post-PCR melting curve was obtained using 15 s at 95 C, 1 min at 60 C, and 15 s at 95 C. The relative fold gene expression was calculated using the 2^−ΔCt^ method [49], and the results were expressed as a percent of control.

### 4.10. Densitometry

For densitometry, immunoblots were scanned using G Box Chemi (Syngene), and the density of each lane of phosphorylated and total protein was quantified, using Image J software. Phosphorylated protein densities were normalised to GAPDH densities, assuming 1 for untreated cells, and then they were converted to a percent of the appropriate control.

### 4.11. Statistical Evaluation

Results are represented as mean ± s.e.m. of at least three independent experiments each performed in triplicate. Statistical analysis was performed using GraphPad Prism 5.0 software (GraphPad Software Inc., San Diego, CA, USA). Significance was determined using ANOVA Dunnett’s Multiple Comparison Test. The statistical significance of differences was indicated in figures by asterisks as follows: * *p* ≤ 0.05, ** *p* ≤ 0.01, and *** *p* ≤ 0.001.

## 5. Conclusions

Our results indicate that CK2-mediated phosphorylation of TS and DHFR may affect their expression and distribution in A-549 cells. The obtained data are in agreement with the previous results indicating the potential regulatory role of CK2-mediated phosphorylation of TS and DHFR. Moreover, our results demonstrate that TS and DHFR can be phosphorylated by not only endogenous CK2 but also likely by other serine/threonine kinase. Our in-gel assay results also indicate that CK2α’ isoform is involved in TS and DHFR phosphorylation; however, this needs in-depth research. Considering the results showing that silencing CK2α increased expression of TS and DHFR mRNA in A-549, the next studies should consider the effect of the protein synthesise inhibitors and mRNA stability regulators on both TS and DHFR expression. Given that protein phosphorylation is an important mechanism for regulating and controlling its activity and function, and as TS together with DHFR are well-established targets in chemotherapy, our research may contribute to understanding the intracellular role of CK2. Furthermore, it may further facilitate the development of effective anti-cancer therapies based on simultaneous inhibition of CK2 or other serine/threonine kinases and enzymes involved in thymidylate synthesis.

## Figures and Tables

**Figure 1 ijms-24-03023-f001:**
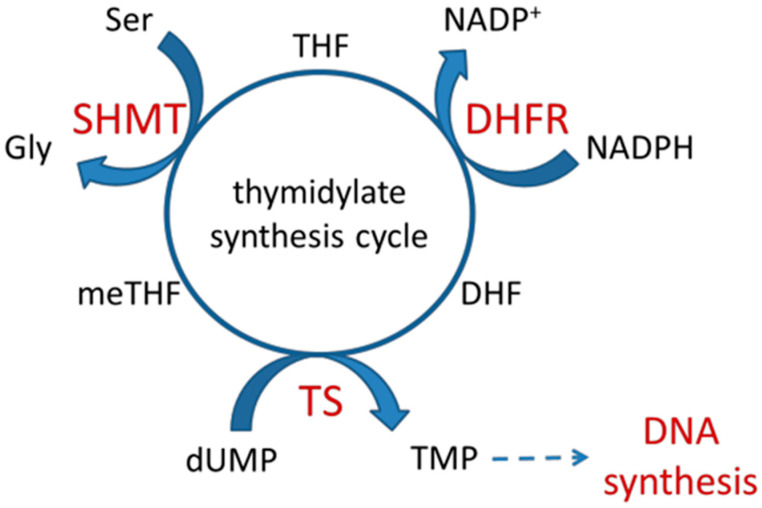
Thymidylate synthesis cycle. TS—thymidylate synthase; DHFR—dihydrofolate reductase; SHMT—serine hydroxymethyltransferase; dUMP—2′-deoxyuridine-5′-monophosphate; TMP, TDP, TTP—2′-deoxythymidine-5′-mono-, di-, and triphosphate, respectively; DHF—dihydrofolate; THF—tetrahydrofolate; meTHF—N5,10-methylenetetrahydrofolate.

**Figure 2 ijms-24-03023-f002:**
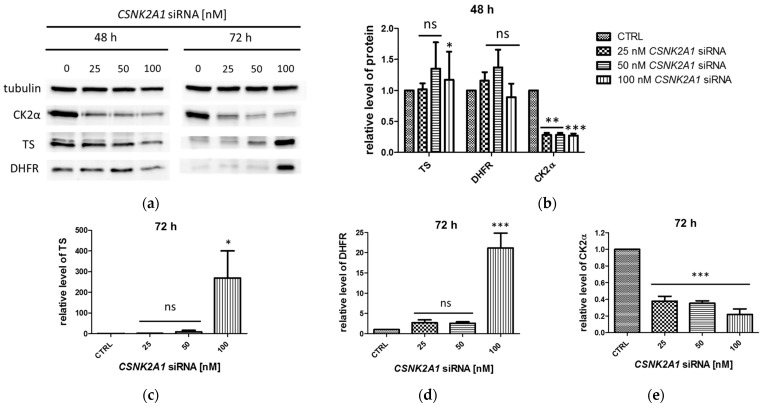
Immunodetection of CK2α, TS, and DHFR proteins in A-549 treated with 25 nM, 50 nM, and 100 nM *CSNK2A1* siRNA for 48 h and 72 h. (**a**) A representative Western blot. (**b**–**e**) Densitometry analysis data for the tested proteins in A-549 treated with *CSNK2A1* siRNA for 48 h (**b**), and for TS (**c**), DHFR (**d**), and CK2α (**e**) after 72 h of treatment with *CSNK2A1* siRNA. Tubulin was used as a loading control for each sample. Section 4 describes the preparation of cell extracts and protein detection. Densitometry quantifications for each tested protein, given under each cell line panel, were calculated with untreated cells (CTRL) as the reference point. Graphs represent mean values ± s.e.m. * *p* < 0.05, ** *p* < 0.01, *** *p* < 0.001, relative to control; ns—not significant.

**Figure 3 ijms-24-03023-f003:**
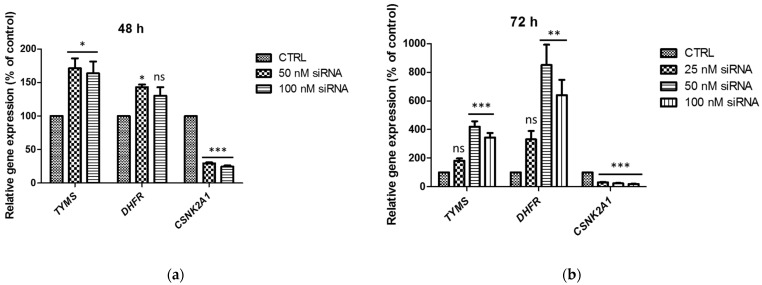
The relative gene expression of thymidylate synthase (*TYMS*) and dihydrofolate reductase (DHFR) in A-549 treated with 50 nM and 100 nM *CSNK2A1* siRNA for 48 h (**a**) and with 25 nM, 50 nM, and 100 nM *CSNK2A1* siRNA for 72 h (**b**). * *p* < 0.05, ** *p* < 0.01, *** *p* < 0.001, relative to control; ns—not significant.

**Figure 4 ijms-24-03023-f004:**
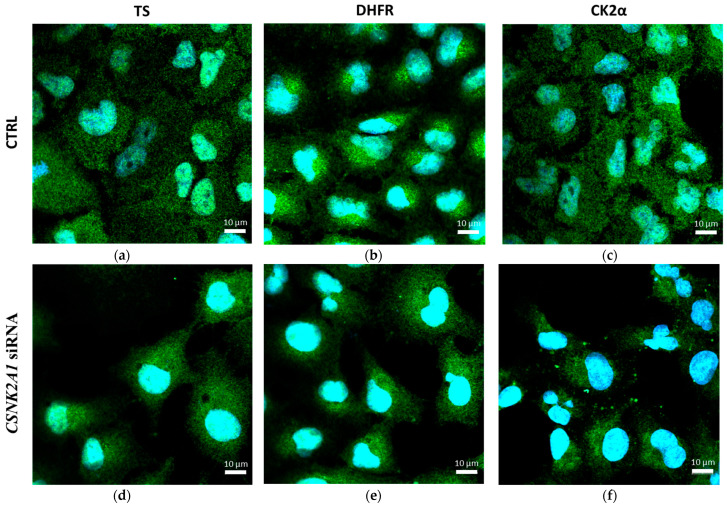
The effect of silencing of *CSNK2A1* on localisation of DHFR (**e**) and TS (**d**) in A-549 cells. Detection of TS, DHFR and CK2α in control cells are shown in (**a**), (**b**), (**c**), respectively. The cells were treated with 100 nM *CSNK2A1* siRNA for 72 h (**d**–**f**), fixed, and after blocking, probed with the primary anti-DHFR or anti-TS antibodies. Subsequently, the cells were treated with Alexa Fluor 488-conjugated anti-mouse secondary antibody (green fluorescence) and Hoechst 33, 342 (nuclei, blue fluorescence).

**Figure 5 ijms-24-03023-f005:**
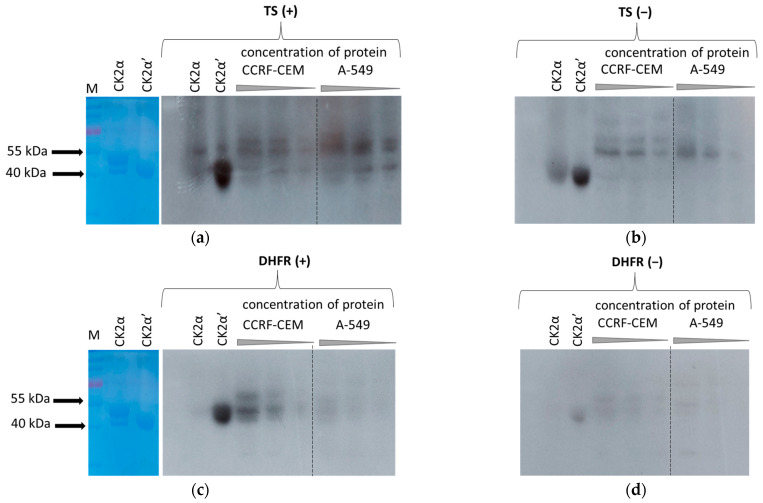
In-gel kinase assay results for TS (**a**) and DHFR (**c**). CCRF-CEM and A-549 lysates were prepared in NP-40 buffer (20 mM Tris-HCl pH 8.0; 137 mM NaCl; 1% Nonidet P-40; 2 mM EDTA). The left parts of (**a**) and (**c**) represent a fragment of gel stained with CBB with visible protein markers (M, PageRuler Prestained Protein Ladder, Thermo Scientific) and recombinant forms of CK2α and CK2α’. Human recombinant catalytic subunits of CK2, i.e., CK2α (29 µg) and CK2α’ (8 µg), were used as positive controls. Lysates containing proteins at three different levels were loaded in the range of 89–22 µg/well. Co-polymerisation of TS and DHFR was at acrylamide:bis-acrylamide ratio 37.5:1 and 19:1, respectively. Autoradiograms shown in (**a**,**b**), and (**c**,**d**) were obtained after 18 h and 6 h of X-ray films exposure, respectively.

**Figure 6 ijms-24-03023-f006:**
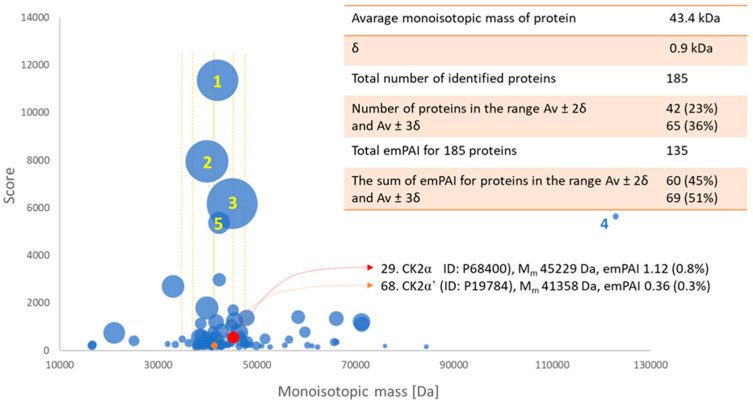
The molecular weight distribution of the proteins in the excised gel band against the Score describes the probability of matching experimental data with proteins in the database. The bubble size corresponds to emPAI value—the parameter dependent on the protein concentration. Most abundant proteins are indicated with numbers: 1. Actin cytoplasmic 1/2, 2. Fructose-bisphosphate aldolase A, 3. Phosphoglycerate kinase 1, 4. POTE ankyrin domain family member E, 5. Actin, alpha cardiac muscle.

## Data Availability

Not applicable.

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
