# Peer review of "Phosphorylation of Thymidylate Synthase and Dihydrofolate Reductase in Cancer Cells and the Effect of CK2α Silencing"

_ijms, 2023, doi:10.3390/ijms24033023_

Round 1
Reviewer 1 Report
In the present study, Patrycja et al demonstrated that knockdown of CK2a increased levels of TS and DHFR at both mRNA and protein levels in cancer cells. They further showed that CK2 might phosphorate both TS and DHFR. This is a fundamental tumor biological study with interesting observations. However, there are several shortcomings to be addressed prior to publication as below:
1. In Figure 2, authors showed that silencing CK2a increased expression of TS and DHFR protein. However, they further displayed that silencing CK2a increased expression of TS and DHFR mRNA in Figure 3. These data indicate that CK2a regulates TS and DHFR in both protein and mRNA levels. They need more data to confirm the regulation of CK2a on TS and DHFR. Protein synthesize inhibitors and mRNA stability regulators should be included in these experiments to further support the conclusion they presently demonstrated.
2. In both Figure 2 and 3, they investigated the effects of CK2a knockdown on TS and DHFR expression. It would be better to shown whether the changes of TS and DHFR expression by CK2a affect cancer cell biological function, such as cell growth and survival and so on.
3. In Figure 4, they demonstrated the effects of CK2a silencing on TS and DHFR localization in cells. The image quality is poor and should be significantly improved. The negative staining controls should be included to clearly observe the localization of TS and DHFR. It would be plus if they can demonstrate the western blot results using cytoplasm and nuclei protein upon CK2a silencing.
4. The current title should be changed. This is because the current data are not only observe phosphorylation of TS and DHFR.
5. Additionally, the term CNK2A1 should be consistent in the text trying not to using both CNK2A1 and CSNK2A1.
Reviewer 2 Report
This manuscript presents novel and relevant data about the question of whether or not the silencing of the CK2 gene impacts the expression of TS and DHFR in A-549 cells. The experimental studies feature excellent designs, implementations, descriptions, and interpretations of the results. The findings add credibility to the conclusions. I think this manuscript is acceptable for publication.
Author Response
We are thankful for all the Reviewer’s comments and suggestions. We have corrected our manuscript according to all the reviewers’ suggestions.
Round 2
Reviewer 1 Report
The authors addressed all of my concerns. This paper will be a nice addition to the field.